# Venous Resection During Pancreatoduodenectomy for Pancreatic Ductal Adenocarcinoma—A Multicentre Propensity Score Matching Analysis of the Recurrence After Whipple’s (RAW) Study [note 1]

**DOI:** 10.3390/cancers17071223

**Published:** 2025-04-04

**Authors:** Ruben Bellotti, Somaiah Aroori, Benno Cardini, Florian Ponholzer, Thomas B. Russell, Peter L. Labib, Stefan Schneeberger, Fabio Ausania, Elizabeth Pando, Keith J. Roberts, Ambareen Kausar, Vasileios K. Mavroeidis, Gabriele Marangoni, Sarah C. Thomasset, Adam E. Frampton, Pavlos Lykoudis, Nassir Alhaboob, Hassaan Bari, Andrew M. Smith, Duncan Spalding, Parthi Srinivasan, Brian R. Davidson, Ricky H. Bhogal, Daniel Croagh, Ismael Dominguez, Rohan Thakkar, Dhanny Gomez, Michael A. Silva, Pierfrancesco Lapolla, Andrea Mingoli, Alberto Porcu, Nehal S. Shah, Zaed Z. R. Hamady, Bilal Al-Sarrieh, Alejandro Serrablo, Manuel Maglione

**Affiliations:** 1Department of Visceral, Transplant and Thoracic Surgery, Medical University of Innsbruck, 6020 Innsbruck, Austria; ruben.bellotti@tirol-kliniken.at (R.B.); benno.cardini@i-med.ac.at (B.C.); florian.ponholzer@i-med.ac.at (F.P.); stefan.schneeberger@i-med.ac.at (S.S.); 2Department of HPB Surgery, University Hospitals Plymouth NHS Trust, Plymouth PL6 8DH, UK; s.aroori@nhs.net (S.A.); peter.labib@nhs.net (P.L.L.); 3Department of HPB Surgery, Hospital Clínic de Barcelona, 08036 Barcelona, Spain; ausania@clinic.cat; 4Department of HPB Surgery, Hospital Universitari Vall d’Hebron, 08036 Barcelona, Spain; elizabeth.pando@vallhebron.cat; 5Department of HPB Surgery, University Hospitals Birmingham NHS Foundation Trust, Birmingham B7 5TE, UK; keith.roberts@uhb.nhs.uk; 6Department of HPB Surgery, East Lancashire Hospitals NHS Trust, Blackburn BB10 2PQ, UK; ambareen.kausar@elht.nhs.uk; 7Department of HPB Surgery, University Hospitals Bristol and Weston NHS Foundation Trust, Bristol BS1 3NU, UK; vasileios.mavroeidis@nhs.net; 8Department of HPB Surgery, The Royal Marsden NHS Foundation Trust, London SW3 6JJ, UK; ricky.bhogal@rmh.nhs.uk; 9Department of HPB Surgery, University Hospital Coventry & Warwickshire, Coventry CV2 2DX, UK; gabriele.marangoni@uhcw.nhs.uk; 10Department of HPB Surgery, NHS Lothian, Edinburgh EH16 4SA, UK; sarah.thomasset@nhslothian.scot.nhs.uk; 11Department of HPB Surgery, Royal Surrey NHS Foundation Trust, Guildford GU2 7XX, UK; aframpton@nhs.net; 12Department of HPB Surgery, Hull University Teaching Hospitals NHS Trust, Hull HU3 2JZ, UK; pavlos.lykoudis@nhs.net; 13Department of HPB Surgery, Ibn Sina Specialized Hospital, Khartoum HGGV +R87, Sudan; nassir_alhaboob@yahoo.com; 14Department of HPB Surgery, Shaukat Khanum Memorial Cancer Hospital, Block R 3 Rd, Lahore 54000, Pakistan; hassaan.bari@yahoo.com; 15Department of HPB Surgery, Leeds Teaching Hospitals NHS Trust, Leeds LS9 7TF, UK; andrewmsmith@nhs.net; 16Department of HPB Surgery, Imperial College Healthcare NHS Trust, London W2 1NY, UK; d.spalding@imperial.ac.uk; 17Department of HPB Surgery, King’s College Hospital NHS Foundation Trust, London SE5 9RS, UK; parthi.srinivasan@nhs.net; 18Department of HPB Surgery, Royal Free London NHS Foundation Trust, London NW3 2QG, UK; b.davidson@ucl.ac.uk; 19Department of HPB Surgery, Monash Medical Centre, Melbourne, VIC 3168, Australia; daniel.croagh@monashhealth.org; 20Department of HPB Surgery, Salvador Zubiran National Institute of Health Sciences and Nutrition, Mexico City 14080, Mexico; ismaeldominguez83@gmail.com; 21Department of HPB Surgery, Newcastle upon Tyne Hospitals NHS Foundation Trust, Newcastle upon Tyne NE7 7DN, UK; rohan.thakkar@nhs.net; 22Department of HPB Surgery, Nottingham University Hospitals NHS Trust, Nottingham NG5 1PB, UK; dhanny.gomez@nuh.nhs.uk; 23Department of HPB Surgery, Oxford University Hospitals NHS Foundation Trust, Oxford OX3 7JH, UK; mikesilva10@gmail.com; 24Department of HPB Surgery, Policlinico Umberto I University Hospital Sapienza, 00161 Rome, Italy; lapolla.1526391@studenti.uniroma1.it (P.L.); andrea.mingoli@uniroma1.it (A.M.); 25Department of HPB Surgery, Azienda Ospedaliero Universitaria di Sassari, 07100 Sassari, Italy; alberto@uniss.it; 26Department of HPB Surgery, Sheffield Teaching Hospitals NHS Foundation Trust, Sheffield S10 2JF, UK; nehal.shah@nhs.net; 27Department of HPB Surgery, University Hospital Southampton NHS Foundation Trust, Southampton SO16 6YD, UK; zaed.hamady@googlemail.com; 28Department of HPB Surgery, Swansea Bay University Health Board, Swansea SA12 7BR, UK; bilal.al-sarireh@wales.nhs.uk; 29Department of HPB Surgery, Hospital Universitario Miguel Servet, 50009 Zaragoza, Spain; aserrablo@salud.aragon.es

**Keywords:** neoadjuvant therapy, mesenteric veins, margins of excision, pancreatic cancer, pancreaticoduodenectomy

## Abstract

Radical surgical resection in the form of pancreatoduodenectomy (PD) remains the only curative-intent treatment option for patients with pancreatic ductal adenocarcinoma (PDAC) of the pancreatic head/uncinate process. Where there is venous involvement, venous resection (VR) may be performed to achieve tumour clearance (R0). This multicentre, retrospective study aimed to compare the oncological outcomes of PDAC patients who underwent PD with venous resection (PDVR) to those who underwent PD without venous resection. By selectively focusing on the resection margin at the superior mesenteric vein (SMV) groove and the resected vein, we found that the need for VR was an independent risk factor for R1 resection. In addition, independent of R status, the PDVR patients had reduced overall survival, reduced disease-free survival, and higher rates of disease recurrence.

## 1. Introduction

Pancreatic ductal adenocarcinoma (PDAC) is characterised by poor overall survival (OS), with a five-year survival rate of 13% [1]. Radical surgical resection remains the only curative-intent treatment option for patients who present with resectable disease. Among this group, up to 40% have at least one positive resection margin on their postoperative histology (R1), and most of these will develop cancer recurrence within 18 to 24 months [2]. It is well known that obtaining an R0 resection gives patients the best possible chance of achieving long-term survival [2]. Venous resection (VR) was introduced to achieve tumour clearance in patients with involved named veins [3,4]. As per the International Study Group for Pancreatic Surgery (ISGPS), VR can be performed in primarily resectable tumours that are in <180° contact with the portal vein (PV) or superior mesenteric vein (SMV) when there is no sign of narrowing/occlusion, as well as in borderline resectable (BR) PDAC (tumour contact ≥ 180° or invasion of the SMV/PV ≤ 180° with bilateral narrowing or occlusion, and not exceeding beyond the inferior border of the duodenum) [5,6].

In BR PDAC, the role of VR following neoadjuvant treatment is well defined [7,8,9,10,11]. Recent studies have shown VR to be associated with improved long-term survival, independent of resection technique [12,13]. However, the oncological benefit of VR for tumours with only partial venous involvement (classified as primarily resectable) remains unclear [5,6]. Despite surgical feasibility, VR is challenging and can be associated with increased morbidity and mortality rates, particularly in cases of extensive venous involvement or where cases are performed in low-volume centres [4,14,15,16,17,18,19,20,21,22,23]. In addition, there remains controversy around the achievability of R0 resection [3,4,15,16,17,18,24,25,26,27,28,29,30,31,32] and the long-term survival benefits of VR, with or without reconstruction [4,15,16,18,20,21,26,27,28,31,32,33].

This study aimed to compare the long-term oncological outcomes (survival and recurrence rates) of PDAC patients who underwent pancreatoduodenectomy (PD) with concomitant venous resection (PDVR) with patients that underwent PD alone. By selectively focusing on the resection margins at the SMV groove and the resected vein [30], this study also attempted to evaluate the impact of VR on local radicality and its impact on disease course by excluding the presence of other positive margins.

## 2. Materials and Methods

### 2.1. Study Design

A propensity-score-matched (PSM) multicentric retrospective study was carried out using data extracted from the Recurrence After Whipple’s (RAW) study (IRAS ID: 280423). Reporting is consistent with the STROBE guidelines [34] for observational research. The study was conducted in accordance with the Declaration of Helsinki and approved by the Ethics Committee of the Medical University of Innsbruck (1287/2021).

### 2.2. Data Collection and Inclusion Criteria

Data were extracted from the RAW study (ClinicalTrials.gov NCT04596865) database. Ethical approval for the RAW study was granted by North West—Greater Manchester South Research Ethics Committee (20/NW/0397) and the study was sponsored by University Hospitals Plymouth NHS Trust. Centres in the UK were invited to participate in the study by email and international centres were invited to join via Twitter (now known as X). Any hepatobiliary centre that had performed PD for PDAC during the research window was eligible to participate, with no prerequisite regarding unit size or volume. Consecutive patients undergoing PD for PDAC, ampullary cancer, or distal cholangiocarcinoma at 29 participating centres in eight countries were screened for eligibility. Patients were included if they underwent PD for a histologically confirmed PDAC between June 2012 and May 2015 (inclusive, three-year research window) and had follow-up data available for a minimum of five years postoperatively (patients who died within five years of PD were included).

For the purposes of this sub-study, we only included patients that underwent PD with or without VR for PDAC, i.e., patients with ampullary or cholangiocarcinoma were excluded. We also excluded patients who had an R1 resection where the involved margin was not the SMV groove or the resected named vein. Patients who underwent an R2 resection or arterial resection were also excluded. R1 resection was defined as microscopic evidence of tumour within 1 mm of a resection margin [35]. R2 (macroscopically visible) resection was defined as a macroscopically incomplete resection at the time of PD (tumour knowingly left in situ) [36]. Our inclusion criteria allowed us to exclusively focus on the SMV groove and the resected named vein, analysing the real impact of VR. The cohort selection criteria are outlined in Figure 1.

Participants were identified through existing departmental databases or by requesting patient lists from histopathology or clinical coding departments. Data were collected from patient records, regional cancer registries, primary care data requests, and regional radiology systems.

The collected data included the following: age, sex, date of diagnosis, comorbidities before the diagnosis of pancreatic cancer, data on preoperative blood tests and biliary stenting, histopathologic tumour characteristics (including size, regional lymph node metastases, and margin resection status according to the Leeds protocol [30]), operative procedures, systemic treatment regimen, date and localisation of any relapse, and date of death or date of last follow-up at five years post-PD. The resection margin status (R) was defined by the distance from the tumour to the margin, and whether the involved margin was directly involved with tumour, node, perineural, and/or lymphovascular invasion. The indication for VR at each institution was based on preoperative imaging and was at the discretion of the operating surgeon. In cases where venous invasion was suspected macroscopically, specimens were sent for frozen section analysis to aid decision making.

As the research period was from 2012 to 2015, TNM staging was performed in accordance with the AJCC Cancer Staging Manual, 7th edition [37]. Postoperative complications were defined as per ISGPS definitions [38,39,40], and major complications were defined as Clavien–Dindo grade ≥ 3a [41].

The primary objective of this study was to determine the impact of PDVR on OS and disease-free survival (DFS). The secondary objectives were to investigate the impact of PDVR on overall morbidity, mortality, and cancer recurrence (compared to the PD-only group).

### 2.3. Statistical Analysis and Propensity Score Matching

To address potential sources of bias within the cohort, we used PSM to compare the groups. We selected age, sex, and lymph node status as variables for the PSM, focusing on the long-term oncological impacts of the study. The PDVR and PD groups were matched at a ratio of 1:2 using the optimal pair matching method to ensure a high match quality. Since the optimal pair matching method minimises the differences in propensity scores, no calliper was necessary. We calculated PSM using logistic regression in R Software (version 4.3.3) with the MatchIt package. Post-matching balance was assessed using Student’s *t*-test or Pearson’s chi-squared test.

Statistical analyses were conducted using IBM SPSS 24 (SPSS Inc., Chicago, IL, USA). Continuous variables are reported as medians (range), and categorical variables as frequencies (percentage). For OS and DFS, 95% confidence intervals (CIs) were reported. Pearson’s chi-squared test and the Kruskal–Wallis test were used to compare categorical and continuous variables, respectively. For multivariable analyses, binary logistic regression was employed to identify independent risk factors for R1 resection, and Cox proportional hazard regression was used to identify survival-related factors. The latter was performed on the results of the univariable analysis, with *p*-values lower than 0.200. Histological grading (G) was excluded from the multivariable survival analyses due to missing data (*n* = 35). Two-sided *p*-values of 0.05 or less were considered statistically significant.

## 3. Results

### 3.1. Baseline Characteristics

Of the 3705 patient records that were originally screened by the collaborating centres, 3270 did not fulfil the selection criteria of the RAW study (Figure 1). A total of 435 patients undergoing PD with or without VR for PDAC were allocated into two groups: PD alone and PD with concomitant venous resection (PDVR). Of the 435 patients, 243 were eligible for the final analysis and these patients were allocated to the PD and PDVR groups in a 2:1 ratio. Out of the 81 patients that had PDVR, 34 (42.0%) had sleeve venous resection with venorrhaphy (SVRV) and 47 (58.0%) had a segment of vein resected with end-to-end reconstruction. Information concerning the type of reconstruction (e.g., use of patches/grafts) was not available (not collected as part of the RAW study).

The patients’ baseline characteristics were similar between both groups. Concerning surgical and TNM staging parameters, no significant differences were observed, while the rates of neoadjuvant therapy were significantly higher in the PDVR group, both before and after PSM (*p* = 0.002 and *p* = 0.032, respectively; Table 1).

### 3.2. Oncological Outcomes

Concerning oncological outcomes, a positive resection margin (R1) at the SMV groove or at the named vein margin was significantly more frequent following PDVR (*p* < 0.001). We found that SVRV (*p* < 0.001) and segmental VR (*p* = 0.034) were independent risk factors for R1 resection upon both univariable and multivariable analysis (*p* < 0.001 and *p* = 0.034, respectively). Further independent risk factors for R1 status at the SMV groove or at the named vein resection margin upon multivariable analysis were pT3 (*p* < 0.001) and pN1 stage (*p* = 0.045). All risk factors for an R1 resection are displayed in Table 2.

Interestingly, irrespective of R status at the SMV groove or at the named vein resection margin, we found no significant difference in recurrence (both systemic and local recurrence) rates between the PDVR and PD groups (no evidence of recurrence: 32.1% vs. 43.2%, respectively; *p* = 0.132; Table 1). However, when considering only systemic relapse, we found a non-significantly higher recurrence rate within the PDVR group (49.5% vs. 37.0%, *p* = 0.065).

In addition, independent of R status, we found higher systemic recurrence rates in the PDVR subgroups (R0: 42.6%, R1: 58.8%) and in PD with R1 status (51.7%) compared to PD with R0 status (33.8%, *p* = 0.034). In comparison, no difference could be observed concerning local recurrence rates (PD + R0: 31.6%, PD + R1: 41.4%, PDVR + R0: 42.6%, PDVR + R1: 44.1%; *p* = 0.351; Appendix A).

The median and five-year OS rates were 21 months (CI95%: 19–26) and 22.2% in the PDVR group, compared to 30 months (CI95%: 23–36) and 32.1% in the PD group (*p* = 0.024 before PSM, Figure 2a; *p* = 0.023 after PSM, Figure 2c). This difference in median OS was even more pronounced when considering only patients undergoing upfront surgery (21 vs. 31 months; *p* = 0.007; Appendix A). The DFS was significantly lower in the PDVR group, with a median of 17 months (CI95%: 13–20) and a five-year DFS rate of 23.0% in the PDVR group, compared to 24 months (CI95%: 15–32) and 38.5% in the PD group (*p* = 0.094 before PSM, Figure 2b; *p* = 0.043 after PSM, Figure 2d). Again, poorer DFS among the PDVR patients was even more evident when only patients who underwent upfront surgery were considered (17 (CI95%: 13–20) vs. 27 (CI95%: 15–38) months; *p* = 0.020; Appendix A).

We also examined the survival outcomes based on the resection status in both the PD and PDVR groups and found that a PD/R0 combination resulted in a significantly longer median OS and DFS (35 (CI95%: 26–343) months and 29 (CI95%: 16–41) months, *p* < 0.001 and *p* = 0.025, respectively), compared to PD/R1 (OS: 16 (CI95%: 6–25) months; DFS: 14 (CI95%: 9–18) months), PDVR/R0 (OS: 23 (CI95%: 13–32) months; DFS: 18 (CI95%: 8–27) months), and PDVR/R1 (OS: 21 (CI95%: 15–26) months; DFS: 17 (CI95%: 14–19) months) (Figure 2e,f). Of note, by comparing just the two PDVR groups, no significant difference was observed between R0 and R1 resections with regard to both OS (*p* = 0.928) and DFS (*p* = 0.558). Analysing only patients undergoing upfront surgery confirmed the differences in OS (*p* < 0.001) and DFS (*p* = 0.020, Appendix A). Of note, no significant differences in the rates of adjuvant therapy were observed between the four groups.

In the multivariable analysis, independent risk factors for shorter OS were pN stage (N1 vs. N0: 21 (CI95%: 18–23) vs. 54 months) and re-laparotomy rate (relaparotomy vs. no relaparotomy: 9 (CI95%: 0–20) vs. 27 (CI95%: 22–31) months, both *p* < 0.001), while VR showed a strong tendency towards poorer survival (*p* = 0.059, Appendix A).

Similar results were found in the multivariable analysis of DFS, where only pN (N1 vs. N0: 16 vs. 46 (CI95%: 13–18) months) stage and re-laparotomy rate (relaparotomy vs. no relaparotomy: 8 (CI95%: 5–10) vs. 21 (CI95%: 16–25) months) were significantly associated with worse outcomes (*p* = 0.044 and *p* = 0.005, respectively).

VR and pT3 stage both showed a tendency towards shorter DFS, but the results did not reach statistical significance (*p* = 0.093 and *p* = 0.092, respectively; Appendix A).

In the multivariable analyses, OS and DFS were not found to correlate with the margin status at the SMV groove, margin status at the named resected vein (*p* = 0.472 and *p* = 0.573, respectively), or with the administration of adjuvant therapy (*p* = 0.751 and *p* = 0.295, respectively; Appendix A, respectively).

### 3.3. Clinical Outcomes

We found no difference in the overall postoperative complication rate between the PD and PDVR groups: Clavien–Dindo grade ≥III complication rate (13.3% vs. 9.9%, *p* = 0.283), intensive care unit stay (9.0% vs. 7.4%, *p* = 0.362), length of stay (12.5 vs. 11.0 days, *p* = 0.131), re-laparotomy rate (5.6% vs. 2.5%, *p* = 0.471), readmission rate (11.0% vs. 8.6%, *p* = 0.869), pancreatic fistula rate (4.8% vs. 2.5%, *p* = 0.786), PV/SMV thrombosis rate (0.6% vs. 3.7%, *p* = 0.075), and 90-day mortality (4.0% vs. 6.2%, *p* = 0.530). Details of all clinical outcomes are displayed in Table 3.

## 4. Discussion

This study supports the observation that the need for PDVR represents an indicator of aggressive disease. Firstly, despite concomitant VR, negative resection margins at the SMV groove or the resected vein were achieved in only 58.0% of cases where PDVR was performed. Secondly, even achieving surgical radicality, PDVR did not improve oncological outcomes. These differences were even more pronounced when only patients who underwent upfront resection were considered. This raises the question of whether intraoperative partial venous involvement in an initially defined resectable PDAC justifies proceeding with upfront surgery. Additionally, it prompts consideration of whether current selection criteria for neoadjuvant treatment should be improved to better capture these patients.

PDVR is considered a valuable curative strategy for PDAC with venous involvement [3,24,33]. However, the oncological benefit of PDVR is still a matter of debate since the literature is characterised by several methodological inhomogeneities [4]; it is currently widely accepted in the context of neoadjuvant treatment [7,8,9,10,11].

This study aimed to improve the evidence base behind the oncological benefit of venous resection by using strict selection criteria and by matching disease stage using PSM. More specifically, cases with concomitant arterial resection and/or any margin involvement other than the SMV groove or the resected named vein were excluded. The resulting 58.8% R0 resection rate with a clearance of >1 mm within the SMV groove or in respect to the resected vein in PDVR is similar to that reported by Ravikumar et al. (63.2%) [16]. However, several studies considering the impact of R status during PDVR do not indicate the specific involvement of the SMV groove or of the resected vein [4,18,29,42], while only one publication considers the SMV groove as the most frequently involved margin [29]. This makes it difficult to compare different studies. Furthermore, the comparison is hampered by the inhomogeneous definition of anatomical resectability [4,16,17,29,42]. In this regard, the resectability definition in our series precedes the ISGPS consensus criteria 2018 [5] or the NCCN criteria 2017 [6]. Therefore, the definition for borderline-resectable (BR) PDAC in this multicentre study is also not uniform. As a consequence, the literature is characterised by a significant bias concerning the indication for upfront surgery versus neoadjuvant therapy, which is the currently preferable treatment for BR-PDAC [7,8,9,10,11].

Another important factor is a possible underreporting of microscopic margin involvement [42]. Our series reviewed histopathological margin status using strict pre-defined RAW study criteria (see Methods). Interestingly, in line with other studies [18,43], any type of VR (sleeve or segmental resection) represents an independent risk factor for R1. Moreover, features of advanced tumour growth, like higher pT and pN stage, correlate with higher positive margin status within the SMV groove, again, similar to published series [43]. In contrast, the administration of neoadjuvant chemotherapy did not impact on margin clearance. However, the number of patients that received neoadjuvant chemotherapy was small. Therefore, these results need to be interpreted with caution.

The higher risk of R1 rates within the SMV groove/resected named vein observed in the PDVR group reflects the need for improved patient selection criteria for neoadjuvant treatment, especially for PDACs with any contact and/or involvement of the porto-mesenteric axis. Whilst PDVR was introduced to provide a better oncological long-term survival, ours, as well as other studies, showed significantly reduced OS compared to PD alone [4,14,15,16,17,18,19,20,21]. Interestingly, studies that showed similar OS for both the PDVR and PD procedures are characterised by generally worse outcomes in the PD group (median OS between 14 and 22 months for PD alone) [25,26,27,28].

Several studies noted that the lower OS associated with PDVR was due to higher postoperative morbidity and mortality rates among venous resection patients [4,16,18,20,21,28,31]. However, in our study and three others [15,26,27], no difference was observed in the overall morbidity and mortality in the PDVR group. Therefore, it is unlikely that the lower OS following PDVR can be attributed to higher morbidity and mortality rates. As well as the VR itself, it seems that complex venous reconstruction techniques, including the use of patches or grafts (IGSPS type III and IV) [3,4,16], or concomitant splenic vein ligation [5], could impair postoperative outcomes. Unfortunately, we were unable to compare our results with these studies, as in the PDVR group the type of venous reconstruction was documented (sleeve vs. segmental resection), but the reconstruction type (direct suture vs. prosthetic or autologous graft) was not. Therefore, it could be argued that although the patients had venous resection, they might not have had radical venous resection requiring reconstruction using either synthetic or prosthetic grafts.

With regard to DFS, we also observed significantly shorter DFS in the PDVR group. This suggests that long-term outcomes are related more closely to the biology of the cancer than the type of surgical intervention. As a consequence, patients undergoing VR are more prone to relapse [32] and have shorter DFS. Indeed, the necessity to perform PDVR depends on the anatomical relation between the tumour and the blood vessels. Initially considered as a mere topographic feature of the tumoral mass [44], increasing evidence suggests that this could also serve as an indicator for biological aggressiveness [45]. Macroscopically undetectable biological aggressiveness might reflect the previously discussed higher rates of R1 margin status after PDVR [4,17,18,29]. One might argue whether upfront surgery is justified in those with PDAC with limited venous contact (therefore staged as resectable). These cases are at risk of an intraoperative decision on VR, which represents an independent risk factor for R1 resection and is associated with higher recurrence rates and lower DFS.

Although we do not have data on the preoperative resectability status of the tumours, the findings of this study suggest that the mere contact of the tumour to the porto-mesenteric axis might represent a crucial risk factor for advanced disease. Besides reduced DFS, we also observed a significantly higher systemic recurrence rate in the PDVR group. Furthermore, similar to Anger et al. [46], we found that the local resection status did not impact on OS and DFS in the PDVR group. This is even more pronounced by the finding that only PD with R0 resection had a significant survival advantage, while PDVR reaching R0 had similar outcomes to PD or PDVR with R1 resection. These clinical observations might also be traced back to the peculiar microanatomy of the SMV groove, which lacks a “buffer” of peripancreatic fat (in contrast to the SMA groove), resulting in direct contact between the vein wall and the pancreatic parenchyma [29]. This anatomical feature, together with the typical infiltrative and dispersive growth pattern of PDAC, is probably of more prognostic significance than surgical technique, and is the reason for more advanced than expected disease necessitating PDVR [29,47,48].

These findings challenge the argument in favour of performing upfront PDVR on patients with resectable tumours in contact with the porto-mesenteric axis. Although PDAC with limited PV/SMV contact is considered resectable [5,6], these tumours are high risk for developing micro-metastases [16,21,31,32] with subsequent recurrence. Therefore, a subclassification of currently defined “resectable” PDACs [5,6] into those with and without contact with the porto-mesenteric axis could be advocated to identify patients who may benefit from neoadjuvant treatment. Currently, neoadjuvant treatment is not recommended in either case [48,49,50,51,52,53,54,55].

Our study has several limitations, including the inherent biases associated with a large, multicentre, retrospective study, even if mitigated through PSM. More specifically, the inclusion of different countries may have significantly impacted surgical outcomes [56]. Additionally, both surgical and oncological outcomes for PD are closely linked to the experience and volume of the centre and the surgical team. The cohort selection process in this study may have led to an underestimation of the actual number of complex resections performed at each centre. This is because certain cases, such as those of arterial resection and VR with R1 margins beyond the considered thresholds, were excluded from the analysis. Nevertheless, the cohort is inherently biased due to its high heterogeneity, which may have influenced our results.

Furthermore, details relating to the criteria used to define resectability, the intraoperative indications for venous resection, and the type of venous resection and reconstruction techniques are missing from the dataset. In addition, the study period itself and the different follow-up modalities in different centres, as well as the use of the seventh edition of the AJCC classification and the need for more data on the type of adjuvant therapy, could have led to an under- or overestimation of the survival outcomes. In addition, the relatively small number of patients with what would now be defined as a “borderline resectable” tumour who received neoadjuvant chemotherapy, as well as the lack of data on the administered protocols and the response after therapy, represent further issues. 

Finally, the missing preoperative CA19-9 values, a relevant parameter to assess biological aggressiveness, is one of the major sources of bias. However, it is important to note that the use of CA19-9 is not without issue. Not all patients express CA19-9 (~5–10% of the population are Lewis-antigen-negative and do not produce CA19-9), which limits its utility in certain individuals. In addition, CA19-9 levels can be elevated in benign conditions such as cholangitis, biliary obstruction, and chronic pancreatitis. Therefore, careful interpretation is required in large series such as ours. However, CA19-9 is currently one of the pivotal parameters for defining borderline resectability status, and levels are useful when considering response to neoadjuvant therapy or cancer recurrence.

Despite these limitations, our study is a large, multicentre study that includes five-year follow-up and survival data on all patients that underwent PD or PDVR. In light of these limitations, applying PSM and the strict selection of patients to understand the oncological impact of PDVR in relation to the specific resection status of the SMV region represents added value. Finally, our data underline the necessity to “super-centralise” those patients needing treatment for PDAC with venous involvement, not only because of the surgical skills required and to mitigate the potential complication burden, but also to address the open question of whether even resectable PDAC with limited venous contact in preoperative imaging could profit from other strategies rather than a surgery-first approach.

## 5. Conclusions

This is the first PSM-based study to examine the oncological impact of PDVR by considering the resection margin of the SMV groove and the resected named vein. We observed higher R1 rates after PDVR compared to PD alone. Additionally, although patients undergoing PDVR did not present a higher complication burden, they displayed poorer OS and DFS, independent of R status. These results suggest that the high rate of tumour recurrence in patients undergoing PDVR relates more to the biological behaviour of the tumour (i.e., the unrecognised presence of systemic disease), rather than to surgical radicality. Our findings suggest PDAC involving the SMV/PV represents a risk factor for systemic spread and support the current recommendation to offer neoadjuvant therapy. Further studies are required to investigate whether patients with PDAC tumours with limited venous contact could benefit from neoadjuvant therapy; these patients are currently offered an upfront surgical resection.

## Figures and Tables

**Figure 1 cancers-17-01223-f001:**
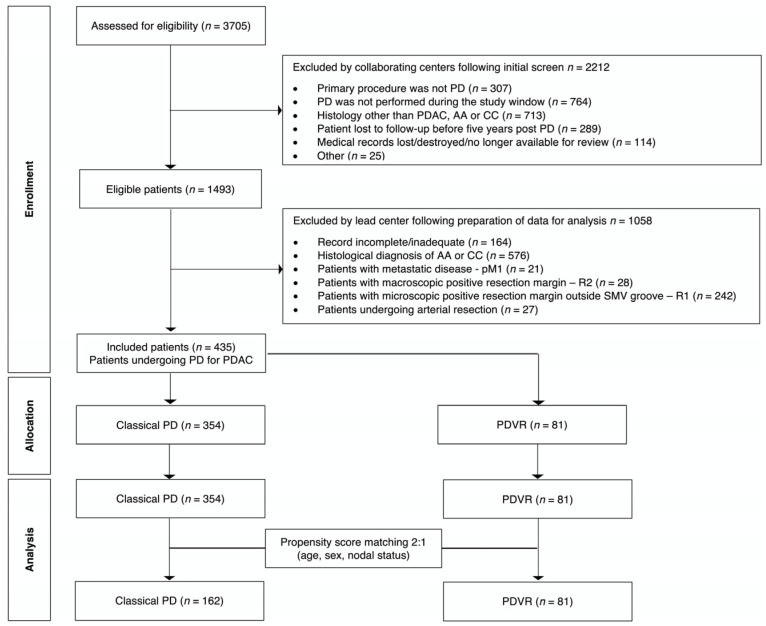
Flow chart for patients’ eligibility and propensity score matching. AA: ampullary adenocarcinoma; CC: cholangiocarcinoma; PD: pancreatoduodenectomy; PDAC: pancreatic ductal adenocarcinoma; PDVR: pancreatoduodenectomy with concomitant venous resection.

**Figure 2 cancers-17-01223-f002:**
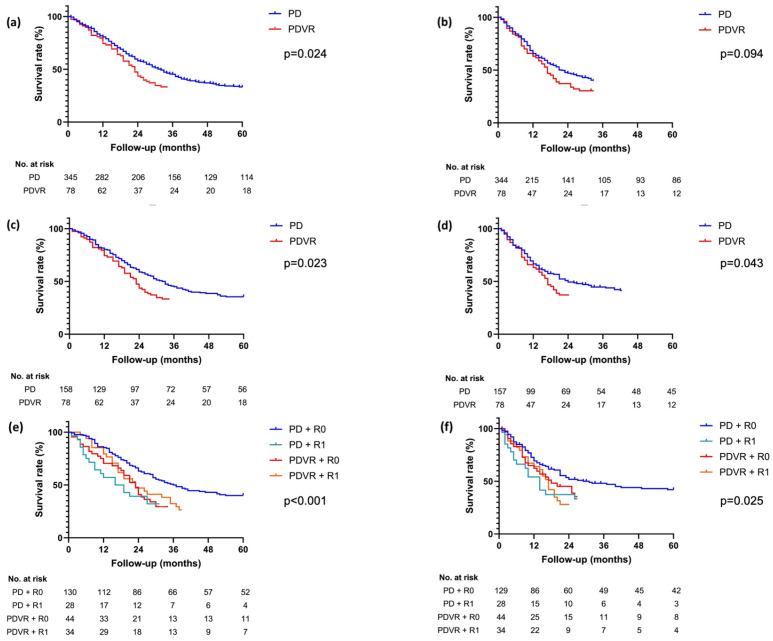
Kaplan–Meier survival curves concerning (**a**) OS and (**b**) DFS before PSM and (**c**) OS and (**d**) DFS after PSM; (**e**) OS and (**f**) DFS considering the combination of the type of resection (PD vs. PDVR) together with the resection margin status (R0 vs. R1).

**Table 1 cancers-17-01223-t001:** Patient baseline characteristics and oncological status.

	Before Propensity Score Matching			After Propensity Score Matching		
Characteristics	PD (*n* = 354) *n* (%)	PDVR (*n* = 81) *n* (%)	*p*	PD (*n* = 162) *n* (%)	PDVR (*n* = 81) *n* (%)	*p*
**Age ≥ 55**	301 (85.0)	68 (84.0)	0.807	136 (84.0)	68 (84.0)	1.000
**Sex ratio (M:F)**	199:155 (56.2:43.8)	44:37 (54.3:45.7)	0.757	88:74 (54.3–45.7)	44:37 (54.3–45.7)	1.000
**Diabetes mellitus**	86 (24.39	22 (27.2)	0.590	39 (24.1)	22 (27.2)	0.601
**Previous malignancies**	54 (15.4)	7 (8.6)	0.114	26 (16.0)	7 (8.6)	0.214
**Respiratory disease**	306 (86.4)	72 (88.9)	0.556	146 (90.1)	72 (88.9)	0.765
**Cardiovascular disease**	149 (42.1)	28 (34.6)	0.214	57 (35.2)	28 (34.6)	0.924
**Neoadjuvant therapy**	16 (4.5)	11 (13.6)	**0.002 ***	9 (5.7)	11 (13.6)	**0.032 ***
**Biliary stenting**	39 (11.0)	7 (8.6)	0.526	10 (6.2)	7 (8.6)	0.477
**Preoperative blood tests ^a^**						
Albumin (g/L)	36.0 (3.0–51.0)	36.0 (2.0–52.0)	0.990	37.0 (3.0–50.0)	36.0 (2.0–52.0)	0.658
Neutrophiles (×10^9^/L)	4.7 (1.0–79.0)	4.9 (2.0–86.0)	0.707	4.7 (2.0–74.0)	4.9 (2.0–86.0)	0.963
Lymphocytes (×10^9^/L)	1.6 (0.0–89.0)	1.6 (1.0–37.0)	0.690	1.7 (0.0–89.0)	1.6 (1.0–37.0)	0.397
Serum bilirubin (μmol/L)	26.0 (1.0–923.0)	24.0 (1.0–383.0)	0.398	20.5 (1.0–636.0)	23.0 (1.0–383.0)	0.787
**ASA ≥ 3**	107 (30.2)	16 (19.8)	0.059	39 (24.1)	16 (19.8)	0.448
**Operation technique**			0.523			0.433
PPPD	186 (53.0)	45 (57.0)		82 (51.6)	45 (57.0)	
Whipple	165 (47.0)	34 (43.0)		77 (48.4)	34 (43.0)	
**Anastomosis technique**			0.252			0.369
PJ	292 (83.4)	72 (86.4)		134 (84.3)	70 (88.6)	
PG	58 (16.6)	9 (13.6)		25 (15.7)	9 (11.4)	
**pT stage (AJCC 7th Edition)**			0.985			0.428
T1	40 (11.3)	9 (11.1)		24 (14.8)	9 (11.1)	
T2	46 (13.0)	10 (12.3)		27 (16.7)	10 (12.3)	
T3	268 (75.7)	62 (76.5)		111 (68.5)	62 (76.5)	
**pN stage (AJCC 7th Edition)**						1.000
N0	110 (31.1)	31 (38.3)	0.212	62 (38.3)	31 (38.3)	
N1	244 (68.9)	50 (61.7)		100 (61.7)	50 (61.7)	
**Grading (*n* = 400)**			0.447			0.188
G1	20 (5.5)	2 (2.6)		12 (8.1)	2 (2.6)	
G2	204 (63.7)	51 (65.4)		90 (60.8)	51 (65.4)	
G3	98 (30.4)	25 (32.1)		46 (31.1)	25 (32.0)	
**Stadium (AJCC 7th Edition)**			0.516			0.517
Ia	27 (7.6)	8 (9.9)		20 (12.3)	8 (9.9)	
Ib	25 (7.1)	5 (6.2)		16 (9.9)	5 (6.2)	
IIa	58 (16.4)	18 (22.2)		26 (16.0)	18 (22.2)	
IIb	244 (68.9)	50 (61.7)		100 (61.7)	50 (61.7)	
**Resection status SMV groove**			**<0.001 ***			**<0.001 ***
R0	275 (77.7)	47 (58.0)		133 (82.1)	47 (58.0)	
R1	79 (22.3)	34 (42.0)		29 (17.9)	34 (42.0)	
**Adjuvant Therapy**	251 (70.9)	61 (75.3)	0.427	118 (72.8)	61 (75.3)	0.680
**Recurrence**			0.272			0.132
None	136 (38.4)	26 (32.1)		70 (43.2)	26 (32.1)	
Local	64 (18.1)	15 (18.5)		32 (19.8)	15 (18.5)	
Systemic	98 (27.7)	20 (24.7)		38 (23.5)	20 (24.7)	
Local + systemic	56 (15.8)	20 (24.7)		22 (13.6)	20 (24.7)	

Values in parentheses are percentages unless indicated otherwise; values are ^a^ median value (95 per cent c.i.); * significant value (*p* ≤ 0.050). Missing data: Previous malignancies (*n* = 0.9 per cent), biliary stenting (*n* = 0.2 per cent), operation technique (*n* = 1.1 per cent), anastomosis technique (*n* = 1.4 per cent), grading (*n* = 9.0 per cent). AJCC: American Joint Committee on Cancer; PG: pancreatogastrostomy; PJ: pancreatojejunostomy; PD: pancreatoduodenectomy; PDVR: pancreatoduodenectomy with concomitant venous resection; PPPD: pylorus-preserving pancreatoduodenectomy.

**Table 2 cancers-17-01223-t002:** Risk factors for R1 resection (after propensity score matching).

	Univariate			Multivariate		
Characteristics	R0 *n* (%)	R1 *n* (%)	*p*	OR	95% CI	*p*
**Age ≥ 55**	150 (83.3)	54 (85.7)	0.697			
**Male sex**	94 (52.2)	38 (60.3)	0.305			
**Diabetes mellitus**	48 (26.7)	13 (20.6)	0.401			
**Previous malignancies**	23 (12.8)	10 (15.9)	0.755			
**Respiratory disease**	159 (88.3)	59 (93.7)	0.335			
**Cardiovascular disease**	61 (33.9)	24 (38.1)	0.645			
**Neoadjuvant therapy**	18 (10.0)	2 (3.2)	0.112	0.373	0.070–1.990	0.248
**Biliary stenting**	13 (7.2)	4 (6.3)	1.000			
**Albumin < 35 g/L ^a^**	65 (39.6)	22 (36.1)	0.625			
**Neutrophiles > 7.5 × 10^9^/L ^a^**	26 (15.0)	16(25.4)	0.065	1.825	0.823–4.048	0.139
**Lymphocytes > 4 × 10^9^/L ^a^**	14 (8.1)	7 (11.1)	0.471			
**Serum Bilirubin > 17 μmol/L ^a^**	93 (51.7)	43 (68.3)	**0.022 ***	1.720	0.862–3.432	0.124
**ASA Scale > 2**	40 (22.2)	15 (23.8)	0.861			
**Operation technique**			0.241			
PPPD	92 (51.1)	38 (60.3)				
Whipple	88 (48.9)	36 (39.7)				
**Venous resection**			**<0.001 ***			
None	133 (73.9)	29 (46.1)				
Wedge	26 (14.4)	21 (33.3)		4.185	1.909–9.176	**<0.001 *** ^$^
Segment	21 (11.7)	13 (20.6)		2.885	1.209–6.885	**0.034 *** ^$^
**pT stage (AJCC 7th Edition)**			**<0.001 ***	3.862	1.459–10.222	**0.007 ***
T1+2	64 (35.6)	6 (9.5)				
T3	116 (64.4)	57 (76.5)				
**pN stage (AJCC 7th Edition)**			**<0.001 ***	2.206	1.017–4.786	**0.045 ***
N0	80 (31.1)	13 (38.3)				
N1	100 (68.9)	50 (61.7)				
**Grading**			0.245			
G1	11 (5.5)	3 (2.6)				
G2	101 (63.7)	40 (65.4)				
G3	52 (30.4)	19 (32.1)				

Values in parentheses are percentages unless indicated otherwise; values are ^a^ median value (95 per cent c.i.); * significant value (*p* ≤ 0.050). Missing data: Previous malignancies (0.4 per cent); grading (*n* = 2.9 per cent). ^$^ corrected accorded to Bonferroni. AJCC: American Joint Committee on Cancer; ASA: American Society of Anaesthesiologists; PPPD: pylorus-preserving pancreatoduodenectomy.

**Table 3 cancers-17-01223-t003:** Postoperative complications.

	Before Propensity Score Matching			After Propensity Score Matching		
Characteristics	PD (*n* = 354) *n*, (%)	PDVR (*n* = 81) *n*, (%)	*p*	PD (*n* = 162) *n*, (%)	PDVR (*n* = 81) *n*, (%)	*p*
CR-POPF	17 (4.8)	2 (2.5)	0.354	5 (3.1)	2 (2.5)	0.786
CR-DGE	30 (8.5)	3 (3.7)	0.143	12 (7.49)	3 (3.7)	0.258
CR-POBL	4 (1.1)	0 (0.0)	0.336	2 (1.2)	0 (0.0)	0.315
CR-POGL	3 (0.8)	1 (1.2)	0.742	0 (0.0)	1 (1.2)	0.156
CR-PPH	26 (7.6)	6 (7.3)	0.984	12 (7.4)	6 (7.4)	1.000
PV thrombosis	2 (0.6)	3 (3.7)	**0.017 ***	1 (0.6)	3 (3.7)	0.075
Clavien–Dindo ≥ 3a	47 (13.3)	8 (9.9)	0.406	24 (14.8)	8 (9.9)	0.283
Postoperative ICU	32 (9.0)	6 (7.4)	0.639	18 (11.1)	6 (7.4)	0.362
Postoperative hospital stay ^a^	12.5 (1–144)	11.0 (2–165)	0.053	12 (4–104)	11 (2–165)	0.131
Re-laparotomy after 30 days	20 (5.6)	2 (2.5)	0.239	7 (4.3)	2 (2.5)	0.471
Readmission after 30 days	39 (11.0)	7 (8.6)	0.531	13 (8.0)	7 (8.6)	0.869
Death within 90 days	14 (4.0)	5 (6.2)	0.378	7 (4.3)	5 (6.2)	0.530

Values in parentheses are percentages unless indicated otherwise; values are ^a^ median value (95 per cent c.i.); * significant value (*p* ≤ 0.050). CR: clinically relevant; DGE: delayed gastric emptying; ICU: intensive care unit; PD: pancreatoduodenectomy; PDVR: pancreatoduodenectomy with concomitant venous resection; POBL: post-operative biliary leakage; POGL: post-operative gastrointestinal leakage; POPF: post-operative pancreatic fistula; PPH: post-pancreatectomy haemorrhage.

## Data Availability

The datasets used and/or analysed during the current study are available from the corresponding author upon reasonable request.

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
