# Peer review of "Venous Resection During Pancreatoduodenectomy for Pancreatic Ductal Adenocarcinoma—A Multicentre Propensity Score Matching Analysis of the Recurrence After Whipple’s (RAW) Studyâ€"

_cancers, 2025, doi:10.3390/cancers17071223_

Round 1
Reviewer 1 Report
Comments and Suggestions for Authors
Ruben Bellotti et al. presented a manuscript titled " Venous Resection during Pancreatoduodenectomy for Pancreatic Ductal Adenocarcinoma – a Multicentric Propensity Score Matching Analysis from the Recurrence After Whipple’s (RAW) Study.” The authors aimed to conduct a multicentric, retrospective study to compare long-term oncological outcomes of patients undergoing pancreatoduodenectomy (PD) with concomitant venous resection (PDVR) against patients who had PD with concomitant venous resection for pancreatic ductal adenocarcinoma (PDAC) from a multi-centre Recurrence After Whipple (RAW) study. The authors concentrated the resection margin in the SMV-groove and on the resected vein and detected the need for VR as a risk factor for R1-resection. The study also intended to evaluate the effect of VR on local radicality and its effect on the disease course. The introduction section of the manuscript covered the fundamental concept of the study undertaken by the authors with supporting literature. The data extracted from the RAW study database and Figure 1 represented the flow chart of patients' eligibility and propensity score matching. The authors determined oncological outcomes and Kaplan-Meier survival curves concerning different cases of patients. All the Tables and figures are presented clearly in the manuscript. This study identified the fact that irrespective of R-status, the PDVR group showed higher systemic recurrence and concluded that patients undergoing PDVR 96 showed lower overall and disease-free survival and higher systemic recurrence rates. The limitations of the study are also discussed by the authors. The presented work is a good piece in comparison to the available reports and encouraging a value addition to the current knowledge. Hence, the manuscript can be accepted after addressing the below-mentioned minor comments:-
- Authors need to discuss any patents published related to the studied subject.
- Future perspectives can be included along with the conclusion.
- Wherever applicable, recent references need to be cited.
Author Response
23rd March 2025
Dear Editor-in-Chief,
Re: Venous Resection during Pancreatoduodenectomy for Pancreatic Ductal Adenocarcinoma – a Multicentric Propensity Score Matching Analysis from the Recurrence After Whipple’s (RAW) Study.
Thank you for considering our manuscript. We are grateful for the comments from your reviewers. Please see our response to the comments below (highlighted in green). We hope you are satisfied with the changes we have made.
Yours sincerely,
Dr. Ruben Bellotti, MD
REVIEWER 1
Ruben Bellotti et al. presented a manuscript titled " Venous Resection during Pancreatoduodenectomy for Pancreatic Ductal Adenocarcinoma – a Multicentric Propensity Score Matching Analysis from the Recurrence After Whipple’s (RAW) Study.” The authors aimed to conduct a multicentric, retrospective study to compare long-term oncological outcomes of patients undergoing pancreatoduodenectomy (PD) with concomitant venous resection (PDVR) against patients who had PD with concomitant venous resection for pancreatic ductal adenocarcinoma (PDAC) from a multi-centre Recurrence After Whipple (RAW) study. The authors concentrated the resection margin in the SMV-groove and on the resected vein and detected the need for VR as a risk factor for R1-resection. The study also intended to evaluate the effect of VR on local radicality and its effect on the disease course. The introduction section of the manuscript covered the fundamental concept of the study undertaken by the authors with supporting literature. The data extracted from the RAW study database and Figure 1 represented the flow chart of patients' eligibility and propensity score matching. The authors determined oncological outcomes and Kaplan-Meier survival curves concerning different cases of patients. All the Tables and figures are presented clearly in the manuscript. This study identified the fact that irrespective of R-status, the PDVR group showed higher systemic recurrence and concluded that patients undergoing PDVR 96 showed lower overall and disease-free survival and higher systemic recurrence rates. The limitations of the study are also discussed by the authors. The presented work is a good piece in comparison to the available reports and encourages a value addition to the current knowledge. Hence, the manuscript can be accepted after addressing the below-mentioned minor comments:
- Authors need to discuss any patents published related to the studied subject.
​To our knowledge, there is no published patent that relates to the studied subject.
- Future perspectives can be included along with the conclusion.
We have included this in the final paragraph.
Lines: 446-449
- Wherever applicable, recent references need to be cited.
Recent evidence concerning VRs for PDAC during PD has now been added.
Lines: 119-121
Lines: 124-125
Reviewer 2 Report
Comments and Suggestions for Authors
This study investigated the prognosis of PDVR and PD patients using the multicenter Recurrence After Whipple's database. The research is well-structured, with complete content, clear logic, and scientific statistical methods. The conclusion that PDVR patients have worse prognosis than PD patients, independent of SMV margin status, reveals the aggressive biological behavior of tumors in PDVR patients, suggesting that these patients might benefit more from neoadjuvant therapy rather than direct surgery. After careful review, we have the following questions and suggestions:
-
Was pathological sampling consistent across all participating centers? Specifically, was Axial slicing (Leeds protocol) used? The R0 resection rate in this study appears unusually high compared to previous studies. If there was no standardized pathological sampling protocol, this should be clearly stated as a study limitation, given the importance of margin status in this research.
-
The high rate of SMV margin positivity in patients undergoing venous resection and reconstruction is difficult to comprehend, as this is not commonly observed in our clinical practice. Could the authors provide further clarification on this finding?
-
The article lacks data on tumor markers (CA19-9, CA125, CEA). Including these would strengthen the study. Since the authors conclude that PDVR group exhibits more aggressive biological behavior, were tumor markers significantly higher in this group? We are particularly interested in whether prognosis differs between PDVR and PD for patients with low tumor markers (e.g., CA19-9 < 100), and whether there's a prognostic difference between R0 and R1 in PDVR patients. The absence of CA19-9 data limits the clinical significance of this study.
-
The main conclusion suggests that upfront PDVR should be avoided for venous-involved resectable tumors in favor of neoadjuvant therapy. However, we believe there might be a logical inconsistency here. If preoperative assessment indicates venous invasion requiring PDVR, these cases should be classified as borderline resectable rather than resectable. For resectable cases (venous contact <180° without morphological changes), if intraoperative findings necessitate PDVR, the option for neoadjuvant therapy would already be unavailable. Therefore, we suggest that a more appropriate study design would compare prognosis between resectable pancreatic cancers with versus without venous contact (as mentioned in the discussion, line 426), rather than PD versus PDVR. We hope the authors will consider this suggestion for their future research on this topic.
Author Response
23rd March 2025
Dear Editor-in-Chief,
Re: Venous Resection during Pancreatoduodenectomy for Pancreatic Ductal Adenocarcinoma – a Multicentric Propensity Score Matching Analysis from the Recurrence After Whipple’s (RAW) Study.
Thank you for considering our manuscript. We are grateful for the comments from your reviewers. Please see our response to the comments below (highlighted in green in the labeled version of the manuscript). We hope you are satisfied with the changes we have made.
Yours sincerely,
Dr. Ruben Bellotti, MD
REVIEWER 2
This study investigated the prognosis of PDVR and PD patients using the multicenter Recurrence After Whipple's database. The research is well-structured, with complete content, clear logic, and scientific statistical methods. The conclusion that PDVR patients have worse prognosis than PD patients, independent of SMV margin status, reveals the aggressive biological behavior of tumors in PDVR patients, suggesting that these patients might benefit more from neoadjuvant therapy rather than direct surgery. After careful review, we have the following questions and suggestions:
- Was pathological sampling consistent across all participating centers? Specifically, was Axial slicing (Leeds protocol) used? The R0 resection rate in this study appears unusually high compared to previous studies. If there was no standardized pathological sampling protocol, this should be clearly stated as a study limitation, given the importance of margin status in this research.
As described in the methods section and in accordance with the reference concerning the further standardisation of the Leeds Protocol on a global scale (Verbeke al., Histopathology, 2021), we confirm the use of a standardised protocol for pathological examination and revision of each specimen, as outlined in the RAW study protocol. We have updated the manuscript to make this clearer.
In our manuscript, we have specifically discussed the R-status within the SMV-groove and/or the resected named vein (we excluded any other R1-margin). In addition, we excluded all cases where concomitant arterial resection was performed. As such, the R0-rates are higher than one might expect.
Lines: 130-131
Lines: 175-177
- The high rate of SMV margin positivity in patients undergoing venous resection and reconstruction is difficult to comprehend, as this is not commonly observed in our clinical practice. Could the authors provide further clarification on this finding?
The higher rate of R1 resection compared to your centre’s experience may reflect the multicentre approach of the RAW study, which included high- and low-volume centres. Argiably, the data is reflective of real-world data. Notably, our data suggests that negative margins of resected vessels do not necessarily translate into better oncological outcomes compared to R1 cases.
- The article lacks data on tumor markers (CA19-9, CA125, CEA). Including these would strengthen the study. Since the authors conclude that PDVR group exhibits more aggressive biological behavior, were tumor markers significantly higher in this group? We are particularly interested in whether prognosis differs between PDVR and PD for patients with low tumor markers (e.g., CA19-9 < 100), and whether there's a prognostic difference between R0 and R1 in PDVR patients. The absence of CA19-9 data limits the clinical significance of this study.
We acknowledge that the lack of CA19-9 data is one of the major limitations of our study. This is outlined in the discussion section. However, it is important to note that the use of CA19-9 as an absolute parameter of biological aggressiveness also has its issues. Not all patients express CA19-9 (~5–10% of the population are Lewis antigen-negative and do not produce CA19-9). This limits its utility in certain individuals. In addition, CA19-9 levels can be elevated in benign conditions such as cholangitis, biliary obstruction, and chronic pancreatitis. Therefore, careful interpretation is required when considering a large case series such as ours. However, CA19-9 is currently one of the pivotal parameters for defining borderline resectability status and levels are useful when considering response to neoadjuvant therapy or cancer recurrence.
In our case series, we observed significant differences when analysing the relationship between pT/pN stage and R-status, and analogously patients undergoing PDVR has higher R1-resection rates, supporting our initial hypothesis of higher aggressiveness in cases of venous contact. However, histological grading as a marker of aggressiveness was not significantly different between R0 and R1 patients. Unfortunately, CA125 and CEA data were also not available to us. We recognise that incorporating tumour markers, molecular pathology and specific gene mutation patterns would be pivotal for future studies focusing on biological tumour behaviour. We have now included an explanation for this in our manuscript.
Lines: 418-426
- The main conclusion suggests that upfront PDVR should be avoided for venous-involved resectable tumors in favor of neoadjuvant therapy. However, we believe there might be a logical inconsistency here. If preoperative assessment indicates venous invasion requiring PDVR, these cases should be classified as borderline resectable rather than resectable. For resectable cases (venous contact <180° without morphological changes), if intraoperative findings necessitate PDVR, the option for neoadjuvant therapy would already be unavailable. Therefore, we suggest that a more appropriate study design would compare prognosis between resectable pancreatic cancers with versus without venous contact (as mentioned in the discussion, line 426-430), rather than PD versus PDVR. We hope the authors will consider this suggestion for their future research on this topic.
We acknowledge the validity of this comment and agree with it. Unfortunately, the patients included in this study were operated before borderline resectability was defined. Thus, patients requiring PDVR in this retrospective study probably include both resectable and borderline resectable cases according to today's definition. Therefore, further studies on the role of neoadjuvant therapy for nowadays defined resectable tumors with limited venous contact are needed. This point was emphasized in the conclusions.
Lines: 446-449
Reviewer 3 Report
Comments and Suggestions for Authors
I have studied carefully the manuscript entitled "Venous Resection during Pancreatoduodenectomy for Pancreatic Ductal Adenocarcinoma – a Multicentric Propensity Score Matching Analysis from the Recurrence After Whipple’s (RAW) Study" by Bellotti R. et al.
The present panuscript deals with an interesting topic, which is awaited to be of interest among the specialized readership. Of note, the authors present the results of their multi-center study by focusing on overall survival and recurrence rates of patients with pancreatic ductal adenocarcinoma (PDAC). These outcomes are compared along two groups of patients: those who underwent pancreatoduodenectomy (PD) with concomitant venous resection (VR), namely PDVR, with those who underwent PD without VR.
The manuscript is well organized. The statistical approach is acceptable. The tables and figures are informative. The language used is of acceptable quality; however, some minor typos and syntax errors could necessitate further professional language editing.
The manuscript is of substantial scientific value. However, the authors are kindly invited to assess/discuss the following issues before considering acceptance.
Major issues
1) The authors are kindly suggested to explicitly determine and discuss the between-groups (between centers) heterogeneity concerning the main outcomes of the study, thus enriching the information provided in the "limitations" paragraph.
Minor issues
1) Line 71: The authors report that "The aim of this multicentric, retrospective study was to compare long-term oncological outcomes of patients undergoing pancreatoduodenectomy (PD) with concomitant venous resection (PDVR) against patients that had PD with concomitant venous resection for PDAC" is rather confusing. The authors are kindly suggested to rephrase.
2) Line 122: The term "RDAC" has been misspelled.
3) Lines 257 - 292: The authors are kindly suggested to provide the 95% confidence intervals for median overall survival (OS) and disease-free survival (DFS).
Comments on the Quality of English LanguageScattered minor typos and syntax errors could necessitate further professional language editing.
Author Response
23rd March 2025
Dear Editor-in-Chief,
Re: Venous Resection during Pancreatoduodenectomy for Pancreatic Ductal Adenocarcinoma – a Multicentric Propensity Score Matching Analysis from the Recurrence After Whipple’s (RAW) Study.
Thank you for considering our manuscript. We are grateful for the comments from your reviewers. Please see our response to the comments below (highlighted in green in the labeled version of the manuscript). We hope you are satisfied with the changes we have made.
Yours sincerely,
Dr. Ruben Bellotti, MD
REVIEWER 3
I have studied carefully the manuscript entitled "Venous Resection during Pancreatoduodenectomy for Pancreatic Ductal Adenocarcinoma – a Multicentric Propensity Score Matching Analysis from the Recurrence After Whipple’s (RAW) Study" by Bellotti R. et al.
The present panuscript deals with an interesting topic, which is awaited to be of interest among the specialized readership. Of note, the authors present the results of their multi-center study by focusing on overall survival and recurrence rates of patients with pancreatic ductal adenocarcinoma (PDAC). These outcomes are compared along two groups of patients: those who underwent pancreatoduodenectomy (PD) with concomitant venous resection (VR), namely PDVR, with those who underwent PD without VR.
The manuscript is well organized. The statistical approach is acceptable. The tables and figures are informative. The language used is of acceptable quality; however, some minor typos and syntax errors could necessitate further professional language editing.
The manuscript is of substantial scientific value. However, the authors are kindly invited to assess/discuss the following issues before considering acceptance.
Major issues
1) The authors are kindly suggested to explicitly determine and discuss the between-groups (between centers) heterogeneity concerning the main outcomes of the study, thus enriching the information provided in the "limitations" paragraph.
We agree with this point entirely. We have now included a more detailed description of these limitations, along with additional references.
Lines: 399-408
Minor issues
1) Line 71: The authors report that "The aim of this multicentric, retrospective study was to compare long-term oncological outcomes of patients undergoing pancreatoduodenectomy (PD) with concomitant venous resection (PDVR) against patients that had PD with concomitant venous resection for PDAC" is rather confusing. The authors are kindly suggested to rephrase.
The sentence has been rephrased.
Lines: 73-75
2) Line 122: The term "RDAC" has been misspelled.
This has been corrected.
Line: 119
3) Lines 257 - 292: The authors are kindly suggested to provide the 95% confidence intervals for median overall survival (OS) and disease-free survival (DFS).
We have completed the data by adding the 95% confidence interval of the requested parameters within the paragraph. Where events occurred very infrequently or not at all (e.g., death or relapse), it was not possible to calculate 95% confidence intervals, so we have not been able to include them.
Line: 205
Lines: 253-284
Line: 492
Reviewer 4 Report
Comments and Suggestions for Authors
I read with interest your paper entitled “Venous Resection during Pancreatoduodenectomy for Pancreatic Ductal Adenocarcinoma – a Multicentric Propensity Score Matching Analysis from the Recurrence After Whipple’s (RAW) Study.”
Here you can find my comments and suggestions to improve the strength of the paper:
-
Introduction
The introduction provides a solid background on the dismal prognosis of PDAC, the importance of achieving R0 resection, and the rationale behind venous resection (VR) during pancreatoduodenectomy. While most relevant studies are cited, there is room to incorporate even more recent evidence regarding neoadjuvant strategies and evolving resectability criteria. -
Research Design
The multicentric, retrospective design using a propensity score matching (PSM) approach is appropriate for addressing the study question. The design is robust in mitigating confounding; however, the inherent limitations of a retrospective analysis and potential inter-center variability remain. -
Methods
The methods are generally well described. Detailed inclusion/exclusion criteria, data collection processes, and statistical techniques (including PSM and multivariable analyses) are provided. That said, some aspects (for example, the exact criteria used intraoperatively for deciding VR and details on reconstruction techniques) are less clear, which could affect reproducibility. -
Results
The results are presented clearly and comprehensively. Tables and Kaplan–Meier survival curves effectively illustrate the differences between PD with and without VR. The reporting of R1 resection rates, survival outcomes, and risk factors is detailed and well organized. -
Conclusions
The conclusions drawn are well supported by the presented data. The authors correctly emphasize that PDVR is associated with a higher risk of R1 resection and worse overall and disease-free survival, and they acknowledge that these findings likely reflect the aggressive tumor biology rather than a failure of surgical technique. -
Originality/Novelty
The study addresses an important clinical question with a unique focus on the impact of VR on margin status within the SMV groove. Although venous resection in PDAC has been examined before, the use of strict anatomical criteria and PSM adds novel insight. -
Significance of Content
Given the ongoing debate over the oncological benefits of VR during PD, the findings have significant implications for surgical decision-making in PDAC, particularly in tailoring neoadjuvant strategies. -
Quality of Presentation
Overall, the manuscript is well written and logically structured. Minor clarifications in the methods (especially regarding surgical technique details) and standardization of terminology would further improve clarity. -
Scientific Soundness
The study is methodologically sound with appropriate use of PSM and multivariable analyses. While the retrospective design is a limitation, the statistical handling of confounders is robust. -
Interest to Readers
The findings will be of high interest to pancreatic surgeons, surgical oncologists, and multidisciplinary teams involved in PDAC care, as they directly impact clinical practice and patient selection. -
Overall Merit
The manuscript makes a valuable contribution to the literature by clarifying the oncological impact of venous resection during PD. Despite inherent limitations, the study’s strengths outweigh its weaknesses.
Suggestions to the Authors
-
Enhance the Introduction:
• Consider integrating additional recent references on neoadjuvant therapy and evolving resectability criteria to further contextualize the rationale for VR in PDAC.
• Clarify how the definitions used in your study relate to current ISGPS or NCCN guidelines. -
Clarify Methodological Details:
• Provide more detailed descriptions of the intraoperative criteria used for decision-making regarding VR.
• Expand on the reconstruction techniques if possible (e.g., clarify whether any prosthetic or autologous grafts were used) since these details might influence outcomes.
• Discuss how potential inter-center variability was managed, given the multicenter design. -
Discuss Limitations More Fully:
• Acknowledge the retrospective design and the potential for selection bias even with PSM.
• Comment on the missing data regarding neoadjuvant therapy administration and preoperative tumor markers (such as CA19-9) that might impact interpretation of oncological outcomes. -
Future Directions:
• Suggest that prospective studies or randomized trials, ideally incorporating standardized criteria for VR, are needed to confirm these findings.
• Highlight the potential role of neoadjuvant therapy in improving outcomes for patients with limited venous contact.
In my view, the study provides important insights into the impact of venous resection on oncological outcomes in PDAC. Addressing the points noted above will strengthen the manuscript and enhance its clarity and clinical relevance.
Author Response
23rd March 2025
Dear Editor-in-Chief,
Re: Venous Resection during Pancreatoduodenectomy for Pancreatic Ductal Adenocarcinoma – a Multicentric Propensity Score Matching Analysis from the Recurrence After Whipple’s (RAW) Study.
Thank you for considering our manuscript. We are grateful for the comments from your reviewers. Please see our response to the comments below (highlighted in green in the labeled version of the manuscript). We hope you are satisfied with the changes we have made.
Yours sincerely,
Dr. Ruben Bellotti, MD
REVIEWER 4
I read with interest your paper entitled “Venous Resection during Pancreatoduodenectomy for Pancreatic Ductal Adenocarcinoma – a Multicentric Propensity Score Matching Analysis from the Recurrence After Whipple’s (RAW) Study.”
Here you can find my comments and suggestions to improve the strength of the paper:
- Introduction
The introduction provides a solid background on the dismal prognosis of PDAC, the importance of achieving R0 resection, and the rationale behind venous resection (VR) during pancreatoduodenectomy. While most relevant studies are cited, there is room to incorporate even more recent evidence regarding neoadjuvant strategies and evolving resectability criteria. - Research Design
The multicentric, retrospective design using a propensity score matching (PSM) approach is appropriate for addressing the study question. The design is robust in mitigating confounding; however, the inherent limitations of a retrospective analysis and potential inter-center variability remain. - Methods
The methods are generally well described. Detailed inclusion/exclusion criteria, data collection processes, and statistical techniques (including PSM and multivariable analyses) are provided. That said, some aspects (for example, the exact criteria used intraoperatively for deciding VR and details on reconstruction techniques) are less clear, which could affect reproducibility. - Results
The results are presented clearly and comprehensively. Tables and Kaplan–Meier survival curves effectively illustrate the differences between PD with and without VR. The reporting of R1 resection rates, survival outcomes, and risk factors is detailed and well organized. - Conclusions
The conclusions drawn are well supported by the presented data. The authors correctly emphasize that PDVR is associated with a higher risk of R1 resection and worse overall and disease-free survival, and they acknowledge that these findings likely reflect the aggressive tumor biology rather than a failure of surgical technique. - Originality/Novelty
The study addresses an important clinical question with a unique focus on the impact of VR on margin status within the SMV groove. Although venous resection in PDAC has been examined before, the use of strict anatomical criteria and PSM adds novel insight. - Significance of Content
Given the ongoing debate over the oncological benefits of VR during PD, the findings have significant implications for surgical decision-making in PDAC, particularly in tailoring neoadjuvant strategies. - Quality of Presentation
Overall, the manuscript is well written and logically structured. Minor clarifications in the methods (especially regarding surgical technique details) and standardization of terminology would further improve clarity. - Scientific Soundness
The study is methodologically sound with appropriate use of PSM and multivariable analyses. While the retrospective design is a limitation, the statistical handling of confounders is robust. - Interest to Readers
The findings will be of high interest to pancreatic surgeons, surgical oncologists, and multidisciplinary teams involved in PDAC care, as they directly impact clinical practice and patient selection. - Overall Merit
The manuscript makes a valuable contribution to the literature by clarifying the oncological impact of venous resection during PD. Despite inherent limitations, the study’s strengths outweigh its weaknesses.
Suggestions to the Authors
- Enhance the Introduction:
• Consider integrating additional recent references on neoadjuvant therapy and evolving resectability criteria to further contextualize the rationale for VR in PDAC.
We have now included further references to emphasise the relevance of VRs during PD for PDAC after neoadjuvant strategies. In particular, we have considered the recent findings which show long-term survival is independent of resection technique.
Lines: 119-121
Clarify how the definitions used in your study relate to current ISGPS or NCCN guidelines.
As reported in the updated limitations paragraph, the date were collected at a time where previous definitions were in use. In relation to the actual definition, we considered the PDVR group as borderline resectable but, unfortunately, we could not confirm this radiologically. The outcomes of the PDVR subgroup also seem to reflect the disease course of a non-neoadjuvant pretreated BR-PDAC. This highlights the validity of the actual guidelines.
- Clarify Methodological Details:
• Provide more detailed descriptions of the intraoperative criteria used for decision-making regarding VR.
A more precise description of the decision-making process behind VR has been added to the method section.
Lines: 181-184
Expand on the reconstruction techniques if possible (e.g., clarify whether any prosthetic or autologous grafts were used) since these details might influence outcomes.
Unfortunately, we are not able to retrieve data concerning the type of reconstruction method used in each case (this was not of primary importance when data was collected for the original RAW study). This is now stated as a major limitation. We will take this into consideration in future studies.
Discuss how potential inter-center variability was managed, given the multicenter design.
As reported in the updated methods section, the pathological assessment was performed in accordance with the standards of the Leeds Protocol (Verbeke al., Histopathology, 2021)( lines: 177). Furthermore, a revision of each specimen occurred as outlined in the RAW study protocol. Also, we specifically considered the R-status within the SMV-groove and/or the resected named vein as carefully indicated in each patient report.
Lines 158-161
Finally, in order to minimise the bias derived from a retrospective design, we performed a PSM considering both patient-related (sex and age), as well as tumour-related, features (lymph node status).
Line: 130-131
Lines: 175-177
- Discuss Limitations More Fully:
• Acknowledge the retrospective design and the potential for selection bias even with PSM.
We agree that the multicentric retrospective nature of the study represents a huge limitation, even if mitigated for through a PSM. We have now discussed this in depth in the limitations paragraph.
Lines: 399-408
Comment on the missing data regarding neoadjuvant therapy administration and preoperative tumor markers (such as CA19-9) that might impact interpretation of oncological outcomes.
Unfortunately, CA125 and CEA data were not available to us (this data was not collected as part of the original RAW study). We recognise that incorporating tumour markers, molecular pathology and specific gene mutation patterns would be a valuable addition for future studies focusing on biological tumor behaviour. As such, we acknowledge that the lack of CA19-9 data is one of the major limitations of our study (as is now stated in the updated discussion section).
However, it is important to note that the use of CA19-9 as an absolute parameter of biological aggressiveness also has its issues. Not all patients express CA19-9 (~5–10% of the population are Lewis antigen-negative and do not produce CA19-9). This limits its utility in certain individuals. In addition, CA19-9 levels can be elevated in benign conditions such as cholangitis, biliary obstruction, and chronic pancreatitis. Therefore, careful interpretation is required when considering a large case series such as ours. However, CA19-9 is currently one of the pivotal parameters for defining borderline resectability status and levels are useful when considering response to neoadjuvant therapy or cancer recurrence.The lack of data concerning neoadjuvant treatment, although representing an important bias, is mitigated by the relatively small number of patients receiving preoperative treatments in the time of the study. Nevertheless, this limitation was discussed in the correspondent section within the discussion.
Lines: 414-426
- Future Directions:
• Suggest that prospective studies or randomized trials, ideally incorporating standardized criteria for VR, are needed to confirm these findings.
As reported in the previous comments for reviewer 1 and 2, we have now added a proper suggestion concerning these specific future perspectives at the end of our manuscript.
Lines: 446-449
- Highlight the potential role of neoadjuvant therapy in improving outcomes for patients with
limited venous contact.
This aspect has now been addressed in the discussion section, as well as the consequent suggestion of future studies on this topic.
Lines: 394-397
In my view, the study provides important insights into the impact of venous resection on oncological outcomes in PDAC. Addressing the points noted above will strengthen the manuscript and enhance its clarity and clinical relevance.